# Impact of Nd Doping on Electronic, Optical, and Magnetic Properties of ZnO: A GGA + U Study

**DOI:** 10.3390/molecules28217416

**Published:** 2023-11-03

**Authors:** Qiao Wu, Gaihui Liu, Huihui Shi, Bohang Zhang, Jing Ning, Tingting Shao, Suqin Xue, Fuchun Zhang

**Affiliations:** 1Network Information Center, Yan’an University, Yan’an 716000, China; wq@yau.edu.cn (Q.W.);; 2School of Physics and Electronic Information, Yan’an University, Yan’an 716000, China

**Keywords:** ZnO, first-principle, magnetic properties, optical properties, rare earth element

## Abstract

The electronic, optical, and magnetic properties of Nd-doped ZnO systems were calculated using the DFT/GGA + U method. According to the results, the Nd dopant causes lattice parameter expansion, negative formation energy, and bandgap narrowing, resulting in the formation of an N-type degenerate semiconductor. Overlapping of the generated impurity and Fermi levels results in a significant trap effect that prevents electron-hole recombination. The absorption spectrum demonstrates a redshift in the visible region, and the intensity increased, leading to enhanced photocatalytic performance. The Nd-doped ZnO system displays ferromagnetic, with FM coupling due to strong spd-f hybridization through magnetic exchange interaction between the Nd-4f state and O-2p, Zn-4s, and Zn-3p states. These findings imply that Nd-doped ZnO may be a promising material for DMS spintronic devices.

## 1. Introduction

The incorporation of magnetic ions in a non-magnetic semiconductor can make it acquire many unique physical properties such as the hysteresis phenomenon, magneto-optical effect, abnormal Hall effect, negative giant magnetoresistive effect, etc. Therefore, it can be made into spin diodes, spin field effect transistors, and other devices, having broad potential applications in information storage, transmission, and processing [1,2]. In 2000, Dietl’s research team found that manganese-doped ZnO and GaN can obtain room-temperature ferromagnetism according to their calculations [3] and predicted that transition metal-doped ZnO semiconductors are dilute magnetic semiconductors with room-temperature ferromagnetism, which laid the foundation for the research of oxide-based dilute magnetic semiconductors. ZnO, a semiconductor with high-exciton binding energy (60 eV) and wide bandgap (3.37 eV), is magnetic owing to intrinsic defects, as well as the doping effect of rare earth elements and 3d transition metals and other magnetic ions and shows significant application value in the fields of information storage and processing [4,5,6]. However, there are drawbacks, such as a low Curie temperature. Additionally, the understanding of magnetic sources and magnetic coupling mechanisms is still inconsistent [7]. The valence electrons of rare earth elements are positioned in the 4f orbital, which is surrounded by the 5s orbital and the 5p orbital, and they have a distinct electronic structure and luminescence properties [8,9]. Because of the shielding effect between the outer orbitals and lack of compensation effect, the influence of crystal field, temperature, and carriers on the 4f orbital electron transition is minimal, and even under conditions of elevated temperature or powerful outer electric fields, the rare earth elements’ optical transition is stable [10]. Thus, the researchers have given a lot of attention to the rare earth-doped ZnO’s physical properties. Xu et al. [11] prepared flower-like nano-ZnO and its La- and Nd-doped modified composites by using a hydrothermal method and studied their photocatalytic properties on reactive dyes. The results showed that the excitation wavelength of 3%La–3%Nd/ZnO was extended to the visible region, and the degradation rate of reactive yellow 145 and reactive red 24 reached more than 80%. In the UV–Vis region, ZnO composites significantly reduced the COD and TOC of printing and dyeing wastewater, indicating that 3%La-3%Nd/ZnO is an excellent photocatalyst. Sharma et al. [12] synthesized Ce-doped ZnO by using chemical precipitation and studied its magnetic and optical properties. Their results showed a significantly reduced bandgap and obvious ferromagnetism after the doping of Ce ions in ZnO, which was caused by the higher orbital momentum of localized 4f electrons in Ce. Wen et al. [13] investigated the Ce-doped ZnO’s magnetic and optical properties employing first-principles calculation and discovered that the doped system’s magnetic properties were related to its configuration. Ag/Co co-doped ZnO’s optoelectronic and ferromagnetic (FM) properties were studied by Liu et al. [14], who discovered that both Ag/Co co-doped ZnO and Ag-doped ZnO exhibited FM properties, while co-doped ZnO presented antiferromagnetic (AFM) properties. Mka et al. [15] performed density functional theory (DFT) calculation on the electronic, optical, and magnetic properties of rare earth (RE = Tm, Yb, Ce) doped with ZnO, and according to the findings, REE doping in ZnO had a considerable effect on its magnetic and photoelectric properties, primarily owing to the presence of 4f electrons and the greatly improved conductivity following doping. Zhang et al. [8] carried out first-principles calculations to predicate the electronic structure and magnetism of rare earth-doped ZnO, and the findings presented that La doping in ZnO can result in a diamagnetic ground state, the La and Ce doping had more stability in contrast with Pr, Nd, and Eu doping. The ground state of Pr, Nd, and Eu dopants at Zn sites was weak AFM, but the ground state of Ce dopants was FM. The electronic, optical, and magnetic properties of ZnO semiconductors not only depend on the growth method and kind of crystal but are also greatly affected by environmental conditions such as temperature, external pressure, humidity, and doping concentration [16,17]. In this work, we focus on the effect of different doping concentrations of Nd on the properties of ZnO.

Based on the above research results, ZnO’s electronic, optical, and magnetic properties can be altered by structural design, impurity doping, and co-doping. The dilute magnetic semiconductors’ magnetic properties are primarily determined by the competition of double- and hyper-exchange interactions, which are mainly affected by the doping geometry. However, it is a challenge to experimentally study doping geometry. DFT is an important tool to reveal the relationship between magnetic properties and doping geometry of dilute magnetic semiconductors [18]. The present study utilizes the GGA + U method for calculating the rare earth Nd-doped ZnO’s electronic, optical, and magnetic properties in order to shed some light on electronic device-related research and development with outstanding magnetic and optical properties.

## 2. Results and Discussion

### 2.1. Analysis of System Structure and Stability

The geometric structural optimizations were carried out for the intrinsic ZnO (i-ZnO). We compared the lattice parameters of i-ZnO by this work to the experimental and DFT structure convergence results (Table 1). It can be seen that agreement between the results of the present calculations and experimental data and other DFT data is very good, showing that our theoretical calculation accuracy is high and the theoretical model and calculation parameters are reliable. The geometric structure of Zn_15_Nd_1_O_16_ and Zn_14_Nd_2_O_16_ supercell models were also optimized, and the Zn_14_Nd_2_O_16_ supercell model included four different configurations. The cell parameters and formation energy are listed in Table 2. Compared with the i-ZnO, the lattice parameters of the a- and c-axes of Zn_15_Nd_1_O_16_ and Zn_14_Nd_2_O_16_ have changed. When the concentration of Nd doping increased, the cell parameters of the doped system became larger, and the volume also increased, indicating a lattice mismatch between Nd^3+^ and ZnO, which is consistent with the experiment [19]. In fact, the ionic radius of Nd^3+^ (0.0995 nm) [20] is bigger in contrast with that of Zn^2+^ (0.074 nm) [21]. The system will have a greater volume after replacing Zn^2+^ with Nd^3+^. In addition, after doping, the repulsion between the excess positive charges of Nd^3+^ is increased. Due to these two factors, the system volume after impurity doping will become greater, and the lattice distortion will occur.

To further verify the doping system’s stability and the intricacy of Nd doping, the formation energy *E*_f_ of the Nd-doped ZnO system was computed with the following equations [23,24]:(1)$$E_{f}=E_{\left(\mathrm{ZnO}+\mathrm{Nd}\right)}-E_{\mathrm{ZnO}}-n_{\mathrm{Nd}}\mu_{\mathrm{Nd}}+n_{\mathrm{Zn}}\mu_{\mathrm{Zn}}$$
where $E_{\left(\mathrm{ZnO}+\mathrm{Nd}\right)}$ and $E_{\mathrm{ZnO}}$ are the total energy of the doped and i-ZnO system containing the same numbers of atoms, $n_{\mathrm{Nd}}$ is the number of Nd atoms, $n_{\mathrm{Zn}}$ is the number of replaced Zn atom, and $\mu_{Nd}$ and $\mu_{Zn}$ are the chemical potentials of Nd and Zn (at T = 0 K), respectively. The calculated formation energies are also summarized in Table 2. The formation energy *E*_f_ of the Nd-doped ZnO system was negative and reduced with the rise of doping concentration, indicating easier doping. We calculated and compared the formation energy of the four different configurations of Zn_14_Nd_2_O_16_. According to the lower formation energy, which is easier to form, we chose the a1 configuration with the lowest formation energy. The electronic structure, optical properties, and magnetic properties were calculated and analyzed for Zn_14_Nd_2_O_16_, which is mainly based on the a1 configuration.

### 2.2. Analysis of Energy Band Structure

Under spin polarization conditions, the structure of the spin-up (SU) and spin-down (SD) energy band in i-ZnO and Nd-doped ZnO systems are illustrated in Figure 1, and Fermi level of 0 eV is represented by the dotted line, $E_{f}$. The high symmetry points of the band structure are located at G (0.0, 0.0, 0.0), A (0.0, 0.0, 0.5), H (−0.333, 0.667, 0.5), K (−0.333, 0.667, 0.0), M (0.0, 0.5, 0.0), and L (0.0, 0.5, 0.5). The i-ZnO band structure diagram is depicted in Figure 1a. Employing the modified GGA + U method, the calculated bandgap value for i-ZnO is 3.34 eV, which is essentially consistent with the experimental finding of 3.3 eV [25]. Therefore, the selected U parameter is reliable. Furthermore, the top of the valence band (VB) and the bottom of the conduction band (CB) are positioned at the G points, and the transition type is G-G, indicating that i-ZnO is a direct bandgap semiconductor. Additionally, the SU and SD bands of the i-ZnO system have the same structure, and there was no occurrence of a spin-splitting phenomenon, demonstrating that the i-ZnO material does not reveal magnetic characteristics.

Figure 1b,c are the diagrams of the SU and SD band structure of Nd-doped ZnO systems under spin polarization conditions. The bandgap values of Zn_15_Nd_1_O_16_ and Zn_14_Nd_2_O_16_ are 3.25 eV and 3.08 eV, respectively, indicating that the bandgap decreases with the increase of Nd doping concentration, also shown by the experimental [19]. This will be conducive to improving the doped system’s optical and electronic transport properties. The VB’s top and the CB’s bottom of Zn_15_Nd_1_O_16_ and Zn_14_Nd_2_O_16_ were positioned at point G, and the transition type was G-G, indicating that Nd-doped ZnO is a direct bandgap semiconductor. In the energy bands of Zn_15_Nd_1_O_16_, a deep donor impurity energy level was generated at 1.69 eV from the CB, and the ionization energy was large. From the density of states (DOS) in Figure 2b,c, the impurity energy level has a primary contribution from Nd-4f electrons. Some of the impurity energy levels coincide with the Fermi level, and the trapping effect is significant. The trap effect leads to the accumulation of non-equilibrium carriers on the impurity energy level [26], which reduces the electron-hole recombination velocity and increases luminous efficiency concurrently. The impurity energy level shifted to lower energy with the increase of Nd doping concentration. In the energy band of Zn_14_Nd_2_O_16_, a shallow main impurity energy level was produced at the VB’s top, and the number of impurity energy levels increased, meaning that the number of electrons that underwent level transitions increased. The photoexcited electrons absorbed lower energy and were transferred from the VB to the impurity energy level, and then absorbed lower energy again and further transferred from the impurity energy level to the CB’s bottom, resulting in increased photocatalytic activity and realized the redshift of the absorption spectrum, which is also reported by the experimental [27,28]. The Fermi level in the band structure of Zn_15_Nd_1_O_16_ and Zn_14_Nd_2_O_16_ systems entered the CB because the Nd atom lost three valence electrons of its 5d and 6s states, whereas the valence state of Zn in ZnO was + 2. Nd introduced excess carriers (electrons) that occupied the CB energy level below the Fermi level. These carriers degenerated, and the doped system became N-type degenerate semiconductors, indicating that the conductivity and the metallicity of the Nd-doped ZnO system are enhanced. In addition, the structures for the SU and SD energy bands of the Nd-doped ZnO system were different, resulting in spin splitting, demonstrating that Nd-doped ZnO is magnetic and possesses electromagnetic transport properties.

### 2.3. Analysis of Density of States

Under spin polarization conditions, the total (TDOS) and partial (PDOS) of i-ZnO and Nd-doped ZnO are shown in Figure 2. The GGA + U method corrected the interaction between electrons in atomic orbitals. The DOS distributions of the Zn-3d state and O-2p state in the VB near the Fermi level were noticeably divided, leading to the broadening of the VB, the weakening of p-d hybridization, and the moving of the O-2p state energy band in the direction of low energy. Therefore, the bandgap was widened and consistent with the experimental results. The deep VB (−15.8 eV, −14.3 eV) had a contribution from O-2s states, and it is strongly localized. The VB (−10.3 eV, −6.9 eV) had a contribution from the Zn-3d state and some O-2p state. Because the Zn-3d state and O-2p state were strongly hybridized, strong Zn-O bonds were created. The upper VB (−6.3 eV, 0 eV) had a contribution from O-2p electrons, while the CB had a contribution from Zn-4s and 3p electrons. Additionally, it was discovered that the Zn-4s state determined the bottom position of the i-ZnO’s CB, and the O-2p state determined the top position of the i-ZnO’s VB. Thus, when the Zn and O atoms combined, electrons of these two states interacted to form a chemical bond, and the O-2p state contributed the majority. Furthermore, the total DOS of SU and SD orbitals were completely symmetric for i-ZnO, and the net spin DOS was zero, indicating that i-ZnO material does not exhibit magnetic characteristics.

Figure 2b,c illustrate the TDOS and PDOS of Nd-doped ZnO. At the top of VB, the repulsion of the anti-bonding Zn-3d state and O-2p state caused the shift of the VB towards the high-energy direction, while the interaction between the bonding Zn-3p state and O-2p state led to the shift of the VB in the direction of low energy. Because the bonding effect was greater in contrast with the anti-bonding effect, the VB moved in the direction of low energy. Concurrently, the CB experiences a more significant reduction, resulting in a narrower bandgap for the Nd-doped ZnO system. Moreover, the Fermi level entered the CB, leading to a band-tail effect, which is conducive to improving the doped system’s optical and electronic transport properties. The doping of Nd introduced strongly localized states in the CB, which were separately contributed by Nd-5d and Nd-4f electrons. The concentration of CB carriers increased, and electrons underwent degeneracy, showing N-type degenerate semiconductor characteristics. The doping of Nd primarily involved the hybridization of Nd-5d and Nd-4f states, which had a significant impact on the DOS at the VB top, forbidden band, and CB bottom of the doped system. The energy region (20 eV, −17.7 eV) had a primary contribution from O-2s, Zn-3d, and Nd-5p electrons, (−14.2 eV, −10.9 eV) had a primary contribution from Zn-3d and O-2p electrons, and (9.6 eV, −3.7 eV) had a primary contribution from Zn-3d and O-2p electrons. The impurity energy level of the forbidden band had a primary contribution from Nd-4f electrons. The CB’s localized state had a primary contribution from the hybridization of Nd-5d and Nd-4f electrons. For Zn_15_Nd_1_O_16_ and Zn_14_Nd_2_O_16_ systems, the TDOS curves of the SU and SD orbitals were asymmetric, so the net spin density of states was not zero. The SU state of the Nd-4f electrons was completely occuelectronspied, while the SD state was empty below the Fermi level. The spin splitting was obvious. The Nd-4f electrons had a net magnetic moment (MM), and the SU and SD orbitals of the O-2p electrons were asymmetric, indicating that Nd-doped ZnO has obvious magnetic characteristics and electromagnetic transport capability.

### 2.4. Analysis of Orbital Charges

The Mulliken population analysis was used to describe the transfer of charge after bonding between atoms [29]. Under spin polarization conditions, the Mulliken charge distribution of i-ZnO and Nd-doped ZnO systems are depicted in Table 3. From the table, in i-ZnO, the Zn atom has a strong capacity to release electrons, resulting in a + 0.93 positive charge, while the O atom has a strong capacity to gain electrons, resulting in −0.93 negative charge, primarily owing to the electronic transfer from the Zn-4s state to the O-2p state. Additionally, in i-ZnO, the number of SU and SD electrons in each orbital of Zn was the same as that of O, indicating that i-ZnO was not magnetic, which is consistent with the band structure analysis finding in Section 2.2 and the state density analysis finding in Section 2.3.

In the Nd-doped ZnO system, the doped atom transforms into a center that is positively charged and has properties of donor impurities as a result of losing electrons. For the Zn_15_Nd_1_O_16_ system, the Nd atom lost electrons, resulting in a + 1.05 positive charge. The Zn atom lost electrons, resulting in a + 0.88 positive charge, while the O atom gained electrons, resulting in a −0.99 negative charge due to the electronic transfer from the Nd-6s and Nd-4f states to Zn-3p and O-2p states. For the Zn_14_Nd_2_O_16_ system, the number of electrons lost by two Nd atoms decreased, resulting in + 1.01 and + 0.87 positive charges in total, the number of electrons lost by Zn atoms increased, resulting in a + 0.92 positive charge, and the number of electrons obtained by O atoms correspondingly decreased, resulting in a −0.88 negative charge. The distribution of O-2s and 2p electrons remained almost unchanged after Nd doping, indicating that the Nd-O chemical bond was relatively stable. The Nd atom has a higher number of positive charges, indicating that the Nd atom contributed more electrons due to the difference between the valence electrons of Nd and that of Zn. Furthermore, the findings revealed that the charge numbers of spin-up and spin-down orbitals of Zn_15_Nd_1_O_16_ and Zn_14_Nd_2_O_16_ systems were different, demonstrating that Nd-doped ZnO materials were magnetic, which is in line with the band structure analysis finding in Section 2.2 and state density analysis finding in Section 2.3.

### 2.5. Analysis of Population Value and Bond Length

The population value and bond length (BL) of i-ZnO and Nd-doped ZnO systems were computed in this section for the purpose of more clearly illustrating the charge distribution, and Table 4 lists the calculation findings [30,31]. In contrast with the population value of i-ZnO’s Zn-O bond, the population values of Zn-O_max_ bonds between Zn and O atoms that are not directly connected to the Nd atom in Nd-doped ZnO system were smaller, and the BL was larger, demonstrating that the covalence was weaker. With the rise of doping concentration, the population value of Zn-O_max_ bonds decreased, and the BL increased, which weakened the covalence. The population value of Zn-O_min_ bonds was almost unchanged. The population values of Nd-O bonds parallel and perpendicular to the *c*-axis increased, and the BLs decreased, which weakened the covalence. This is consistent with the analysis outcomes of the lattice constant of the doped system in Section 2.1 and indicates that the doped system’s lattice had distortion, and the centers of positive and negative charge were destroyed to produce a local potential difference that hindered hole–electron pair recombination and improved the photocatalytic performance of ZnO materials. This can be attributed to the fact that the doping of Nd atoms intensified the overlapping of electron shells between atoms, which accelerated electron transfer and enhanced the covalence. Further analysis showed that since the electronegativity of the doped Nd atom is lower than that of the Zn atom, it is easier for the O atom to obtain electrons from the Nd atom. Therefore, the population of the Nd-O bond in the doped system is lower, and the polarity of the Nd-O bond is stronger compared with the Zn-O bond.

### 2.6. Analysis of Differential Charge Density

Under the spin polarization condition, the electron charge density distribution of Nd^3+^ in Zn_15_Nd_1_O_16_ and Zn_14_Nd_2_O_16_ was calculated, which reveals the influence of Nd doping on the electron transport performance of ZnO (Figure 3a,b). The yellow part represents electron depletion, and the blue part represents electron accumulation. The electron charge density distribution showed that the non-uniform charge distributions were intensified after Nd-doped ZnO. Figure 3a,b show that the depletion of electrons is near the Nd, while the electrons accumulated in O near Nd atoms and the nearby Zn atoms are a little disturbed, which is in consistency with the electronegativity principle that the electronegativity of the doped Nd atom is lower than that of the Zn atom, and it is easier for the O atom to obtain electrons from the Nd atom. In particular, the interaction between the Nd atoms forms an Nd-Nd ionic bond in Zn_14_Nd_2_O_16_, which further facilitates electron transfer.

At the same time, the 2D differential charge cross-section was drawn based on differential charge density. The best plane was selected by the atoms marked in Figure 3a,b to calculate. As shown in Figure 3c,d, the scale bar’s red and blue colors represent electron depletion and accumulation, respectively. The darker the color, the more electron transfer. Figure 3c,d show that the doping of Nd^3+^ ions forms an Nd-O ionic bond in Zn_15_Nd_1_O_16_ and forms Nd-O and Nd-Nd ionic bonds in Zn_14_Nd_2_O_16_ with different strengths, revealing a strong electron transfer ability. The result showed that the photogenerated electrons of ZnO were transferred to Nd sites, which inhibits the recombination of electron–hole pairs.

### 2.7. Optical Properties

The semiconductors’ optical properties have a strong link to the electronic structure. The bulk materials’ optical properties are typically examined in the linear response region employing the dielectric functions (DFs). Thus, the DFs, absorption coefficients, reflectivities, and energy loss functions (ELFs) were calculated to systematically analyze the influence of the doping of Nd on the optical properties of ZnO. In accordance with the description of direct transition probability and Kramers–Kronig dispersion relation, the imaginary part $\varepsilon_{2}(\omega )$ and real part $\varepsilon_{1}(\omega )$ of the crystal DF, absorption coefficient, reflectivity, and ELF can be derived, and the results are presented below (see refs. [32,33,34] for detailed derivation process):(2)$$\varepsilon \left(\omega \right)=\varepsilon_{1}\left(\omega \right)+i\varepsilon_{2}\left(\omega \right)$$
(3)$$\varepsilon_{2}\left(\omega \right)=\frac{\pi }{\varepsilon_{0}}{\left(\frac{e}{m\omega }\right)}^{2}\cdot \sum_{V,C}\left\{\int_{BZ}\frac{2d\vec{K}}{{\left(2\pi \right)}^{3}}{\left|\vec{a}\cdot {\vec{M}}_{V,C}\right|}^{2}\delta \left[E_{C}\left(\vec{K}\right)-E_{V}\left(\vec{K}\right)-\hbar \omega \right]\right\}$$
(4)$$\begin{array}{c} \varepsilon_{1}(\omega )=1+\frac{2e}{\varepsilon_{0}m^{2}}\cdot \sum_{V,C}\int_{BZ}\frac{2d\vec{K}}{{\left(2\pi \right)}^{3}}\frac{{\left|\vec{a}\cdot {\vec{M}}_{V,C}\left(\vec{K}\right)\right|}^{2}}{\left[E_{C}\left(\vec{K}\right)-E_{V}\left(\vec{K}\right)\right]/\hbar }\cdot \\ \frac{1}{{\left[E_{C}\left(\vec{K}\right)-E_{V}\left(\vec{K}\right)\right]}^{2}/\hbar^{2}-\omega^{2}} \end{array}$$
(5)$$R\left(\omega \right)=\frac{{\left(n-1\right)}^{2}+k^{2}}{{\left(n+1\right)}^{2}+k^{2}}$$
(6)$$\alpha \equiv \frac{2\omega \kappa }{c}=\frac{4\pi \kappa }{\lambda_{0}}$$
(7)$$L\left(\omega \right)=Im\left(\frac{-1}{\varepsilon \left(\omega \right)}\right)-\frac{\varepsilon_{2}^{2}\left(\omega \right)}{\varepsilon_{1}^{2}\left(\omega \right)+\varepsilon_{2}^{2}\left(\omega \right)}$$
where $\varepsilon_{0}$ is the vacuum’s dielectric constant, $\lambda_{0}$ is the vacuum’s wavelength, ℏ is the Planck constant, C and V are the CB and VB, respectively, BZ denotes the first Brillouin zone, $\vec{K}$ is the electron wave vector, a is the unit direction vector of the vector potential A, $M_{V,C}$ is the transition matrix’s element, ω is the electromagnetic frequency, R(ω) is the reflectivity, $E_{C}\left(K\right)$ and $E_{V}\left(K\right)$ are the intrinsic energy level of the CBs and VBs, respectively, L(ω) is the ELF, and α is the absorption coefficient.

The DF curves of i-ZnO and Nd-doped ZnO systems are depicted in Figure 4. Figure 4a presents the curves of the DF’s real part. The average static dielectric constant of i-ZnO, Zn_15_Nd_1_O_16_, and Zn_14_Nd2O_16_ were 3.21, 5.58, and 10.70, respectively. In contrast with i-ZnO, the dielectric constants of these doped systems all increased, which is consistent with the previous experiment [28], indicating that Nd doping enhanced the polarization of the system and prolonged the lifetime of photoelectrons in the CB, indicating that Nd doping improved the photocatalytic performance of the system. With the rise of the doping concentration, the photocatalytic performance was enhanced. The average static dielectric constant of the Zn_14_Nd_2_O_16_ system was the largest, demonstrating that its photocatalytic activity was the best.

The curves of the DF’s imaginary part are depicted in Figure 4b. Four dielectric peaks were visible in the imaginary part of the i-ZnO’s DF. The first peak positioned at 5.82 eV was primarily caused by electronic transitions from the O-2p state at the VB’s top to the Zn-4s state at the CB’s bottom. The second peak positioned at 9.15 eV was primarily owing to electronic transitions from the Zn-3d and O-2s states in the VB far away from the Fermi surface to the O-2p and Zn-3d states at the VB’s top. The third peak positioned at 13.33 eV was primarily derived from the electronic transitions from the Zn-3d, O-2s, and O-2p states in the VB away from the Fermi surface to the Zn-4s state at the CB’s bottom. The fourth peak positioned at 15.71 eV was primarily attributed to the electronic transitions from the O-2s, Zn-3d, and Zn-4s states in the VB far away from the Fermi surface to the O-2p state at the VB’s top. After Nd doping, a new peak near 0.7 eV emerged, primarily owing to the electronic transitions from the impurity levels in the bandgap to the CB’s bottom. According to the PDOS in Figure 2, in the Nd-doped ZnO system, the first peak was primarily produced by the electronic transitions from the impurity level of the Nd-4f state coinciding with the Fermi surface to the Zn-4s state at the CB’s bottom. With the increase in doping concentration, the peak value and the transition probability increased. Compared with i-ZnO, the peaks around 5.82 eV and 9.15 eV of Nd-doped ZnO systems were shifted towards the high energy direction (blueshift), with an increased peak value. The peaks near 13.33 eV and 15.71 eV were shifted towards the low energy direction (redshift), with a decreased peak value.

The absorption and reflection spectra of i-ZnO and Nd-doped ZnO systems are presented in Figure 5. Figure 5a displays the absorption spectra that were calculated using Equation (6). The absorption coefficient was 10^5^ cm^−1^, and the absorption edge of i-ZnO located around 3.30 eV denoted the onset of intrinsic absorption, which investigates the important ZnO’s optical properties. The absorption coefficient undergoes a sharp enhancement in the order of 10^5^ with the rise of photon energy, meaning strong optical absorption occurred. The direct transition began, and the absorption edge of Nd-doped ZnO moved towards the high energy direction, resulting in a blueshift. This was induced by the introduction of several carriers (electrons) by donor atoms of Nd. It was in agreement with the experimental results that a remarkable absorption band shifted toward the longer wavelength region with the increase of Nd doping concentration [27,28]. In addition, the Fermi level entered the CB, leading to the Burstein–Moss effect [33,35]. In the visible region (1.63 eV–3.10 eV), the i-ZnO’s optical absorption was extremely marginal and almost non-existent, while the Nd-doped ZnO system’s optical absorption was significantly enhanced. This was ascribed to the formation of a deep impurity energy level partially coinciding with the Fermi level in the forbidden band after doping, which significantly improved the photocatalytic performance of ZnO. The optical absorption intensity of Zn_14_Nd_2_O_16_ was the greatest, signifying that the photocatalytic activity of this system was the best. The result was consistent with the previous experiment, which showed that the photocatalytic activity of ZnO increased via Nd doping [28]. The maximum optical absorption intensity was achieved at around 16.7 eV. Compared with i-ZnO, the maximum light absorption peak was shifted towards the low energy direction, resulting in an obvious redshift. As the doping concentration increased, the redshift phenomenon became more pronounced, expanding the infrared absorption range of ZnO. On the basis of the PDOS diagram in Figure 2, the maximum absorption peak is produced by electronic transitions from the O-2s state in the deep VB to the CB’s bottom.

Figure 5b shows the curves for the reflectivity of i-ZnO and Nd-doped ZnO systems, which were determined using Equation (5). The static reflectivity R (0) of i-ZnO was 0.08, while R (0) of Zn_15_Nd_1_O_16_ and Zn_14_Nd_2_O_16_ were 0.17 and 0.29, respectively. In the range of 16 eV to 20 eV, the reflectivity was high. For i-ZnO, the highest reflectivity of 0.24 was reached at 17.2 eV. The highest reflectivity was 0.25 at 17.4 eV for Zn_15_Nd_1_O_16_ and 0.22 at 16.5 eV for Zn_14_Nd_2_O_16_. The maximum reflectivity of i-ZnO and Nd-doped ZnO systems appeared in the ultraviolet region, and the magnitude was approximately 0.2. With the rise of Nd doping concentration, the reflectivity peak decreased; the reflectivity peak of the single-Nd-doped ZnO system was larger than that of i-ZnO, and the peak underwent a blueshift, and the reflectivity peak of dual-Nd-doped ZnO system was less than that of i-ZnO, and the peak underwent a redshift. In the low-energy region, the reflectivity of the Nd-doped system was higher than that of i-ZnO. Some photons are reflected, some are absorbed, and the remaining portion is transmitted through the substance as they travel through it. Therefore, the absorption coefficient and reflectivity increased in the region of low energy for Nd-doped ZnO and decreased the transmittance. However, the absorption coefficient and reflectivity decreased in the high-energy region, while the transmittance increased in the ultraviolet region.

Figure 6a shows the PC for i-ZnO and Nd-doped ZnO systems, corresponding to the imaginary part $\varepsilon_{2}(\omega )$ of DFs (Figure 4b). In the range of 10 eV to 20 eV, i-ZnO and Nd-doped ZnO systems showed high conductivity, and the PC reached its maximum value near 15 eV. The peak values of i-ZnO, Zn_15_Nd_1_O_16_, and Zn_14_Nd_2_O_16_ were 7.71 fs^−1^, 6.47 fs^−1^, and 5.72 fs^−1^, respectively. In the visible region, the PC of i-ZnO and Nd-doped ZnO systems increased with the rise of the energy of incident photons. The PC of the Nd-doped ZnO system was larger compared to that of i-ZnO, comparing the peak PC of all systems in the low-energy region, which was owing to the increase of the electron density of Nd-4f state at the doped system’s Fermi level, resulting in the increase of the PC in the region of low energy. However, the PC of all systems was small, and as the number of Nd atoms increased, the peak was shifted to the left, indicating that Nd doping improved the PC of ZnO semiconductors in the visible region.

Figure 6b displays the ELF of i-ZnO and Nd-doped ZnO systems. The plasma frequencies of i-ZnO, Zn_15_Nd_1_O_16_, and Zn_14_Nd_2_O_16_ were 40.29 eV, 39.10 eV, and 36.08 eV, respectively. With the rise of Nd doping concentration, the peak value was redshifted, and the intensity was weaker than that of the i-ZnO system.

### 2.8. Magnetic Properties

Rare earth elements have large MM and strong orbital anisotropy and can possibly improve the magnetic properties of doped ZnO semiconductors. According to Dhar et al. [36], Gd-doped GaN has a huge MM, and it is envisaged that the coupling between the 4f electrons of rare earth ions and the host’s electrons can produce stable ferromagnetism, which was conducive to the study of ZnO magnetism. ZnO consists of a large number of s electrons and presents N-type conductivity. By using the spin polarization computational method, we calculated the magnetic coupling and total MM of the intrinsic and Nd-doped ZnO systems (Table 5), as well as the atomic MM and orbital MM (Table 6). Analysis shows that the total MM of i-ZnO was equal to 0 μB, indicating that the Zn_16_O_16_ system was non-magnetic, which was consistent with the band structure analysis finding in Section 2.2 and the state density analysis result in Section 2.3. The result was consistent with the previous report, which showed that pure ZnO samples exhibited diamagnetic nature at room temperature [19]. The sum of the absolute values of MM of the Nd-doped ZnO system was not equal to zero, demonstrating that the doped system was magnetic, also shown by the experiment [19]. The experimental model observed that weak ferromagnetism increases with increasing Nd concentrations in ZnO when the doping concentration is less than 6%, and the weak ferromagnetism decreases when the doping concentration is up to 10% because the Nd carbonate phase increased [27]. Therefore, we designed Zn_15_Nd_1_O_16_ (2 × 2 × 2) supercell and a Zn_14_Nd_2_O_16_ (2 × 2 × 2) supercell which have the appropriate concentration of Nd-doped ZnO in the present study. The total MM of Zn_15_Nd_1_O_16_ and Zn_14_Nd_2_O_16_ systems were 3.75 μB and 5.97 μB, respectively, and with the rise of concentration of Nd doping, the total MM of the doped system increased, showing obvious ferromagnetism.

From Table 6, the atomic MM of the Nd atom was 3.79 µB, the orbital MM of Nd-4f, Nd-4d, and Nd-5s orbitals were 3.67 µB, 0.07 µB, and 0.06 µB, respectively, the atomic MM of surrounding Zn atoms was about 0.01 µB, the orbital MM of Zn-4s orbital was about 0.01 µB, the atomic MM of surrounding O atoms was about −0.03 µB, and the orbital MM of O-2s and O-2p orbitals was about −0.01 µB and −0.02 µB, respectively; in Zn_14_Nd_2_O_16_, the atomic MM of two Nd atoms were 3.06 µB and 3.08 µB, respectively, the orbital MM of Nd-4f and Nd-4d orbitals were 2.96 µB and 0.07 µB, respectively, the atomic MM with a single surrounding Zn atom was about −0.01 µB, the orbital MM of Zn-4s orbital was about −0.01 µB, the atomic MM of the O atom was about −0.02 µB, and the orbital MM of O-2s and O-2p orbitals were about −0.01 µB and −0.01 µB, respectively. We concluded that the total MM of Zn_15_Nd_1_O_16_ has a primary contribution from Nd-4f, Nd-4d, and Nd-5s orbitals, as well as O-2s, O-2p, and Zn-4s orbitals near Nd atoms, while the magnetism of Zn_14_Nd_2_O_16_ primarily came from the coupling between two Nd atoms and O atoms, wherein Nd-4f orbital contributes the most. Additionally, it was found that the coupling between Nd and Zn was FM, and the coupling between Nd and O was AFM.

The above conclusions indicate that the magnetism of the Nd-doped ZnO system originated from the double exchange mechanism caused by orbital–spin interaction [37,38,39], that is, the anisotropic exchange of Nd atoms. In addition, according to the Goodenough–Kanamori rule, because the Nd-4f orbital is in a semi-full state, its ferromagnetism is significant [40,41], and the electrons of Nd-4f orbital are localized electrons that contribute to magnetism, and these localized electrons couple with the conductive electrons in surrounding orbitals such as Zn-3p, O-2s, and O-2p orbitals, and conductive electrons are spin-polarized, resulting in different densities of the SU and SD electrons, thus the spin polarization direction of electrons in Nd-4f orbital is determined and spin transport is realized. Therefore, Nd-doped ZnO systems show FM characteristics, which can be further verified by the partial-wave density analysis of the doped system.

The diagram of TDOS in Figure 2 was analyzed to reveal the Nd-doping effect on the ZnO’s magnetic properties more intuitively. Figure 2a shows that the value of the net DOS of i-ZnO is 0, and the curves of the SU and SD TDOS of i-ZnO are absolutely symmetrical, demonstrating that i-ZnO is non-magnetic. Figure 2b,c present that the values of the DOS of Zn1_5_Nd_1_O_16_ and Zn_14_Nd_2_O_16_ systems are not zero, indicating that the curves of the SU and SD TDOS of the Nd-doped ZnO systems are not completely symmetrical. There was a variation in the number of electrons in the SU and SD directions, demonstrating that the doped system had a net MM, which is consistent with the analysis findings of the total MM. Further analysis of the PDOS in Figure 7 shows that the magnetism of Zn_15_Nd_1_O_16_ and Zn_14_Nd_2_O_16_ is derived from the spin exchange of Nd-4f, Nd-4d, Zn-4s, and Zn-3p states at the CB’s bottom, and the spin exchange of Nd-4f, Nd-4d, and O-2p states at the VB’s top and was caused by the strong hybridization of Nd-4f, Nd-4d, O-2p, Zn-4s, and Zn-3p states near Fermi surface. Among them, the Nd-4f state showed the most notable spin-splitting phenomenon. With the increase of Nd doping concentration, the spd-f hybridization became more intense, and the spin polarization phenomenon became more pronounced, indicating that the net MM was larger.

Because Zn_15_Nd_1_O_16_ and Zn_14_Nd_2_O_16_ systems all exhibited ferromagnetism, indicating that the electrons in such systems underwent spin polarization. Spin polarizability p is generally considered the variation between the DOS of majority carriers (N↑) and the normalized DOS of minority carriers (N↓) at the Fermi surface [42].
(8)$$p=\frac{N↑-N↓}{N↑+N↓}$$

The spin polarizability and the magnetization strength M have the following relationship,
(9)$$M=\mu_{B}\int \left(N↑-N↓\right)dE$$
where *µ*_B_ is the Bohr magneton.

Then, we have
(10)p∝M

Further, in the TDOS near the Fermi surface in Figure 7, the number of majority carriers N↑ at the Fermi surface in Zn_15_Nd_1_O_16_ and Zn_14_Nd_2_O_16_ > 0, while that of minority carriers N↓ was almost 0, so the spin polarizability was close to 100%. The number of majority carriers N↑ above the Fermi surface in Zn_14_Nd_2_O_16_ > 0, while that of minority carriers N↓ was close to 0, so the spin polarizability was less than 100%, indicating that Nd-doped ZnO systems had dilute magnetic semiconductor (DMS) properties.

The net spin density distributions of Zn_15_Nd_1_O_16_ and Zn_14_Nd_2_O_16_ systems were computed, and the findings are presented in Figure 8. The blue signifies the positive spin-charge density, and the yellow signifies the negative spin-charge density, with a unit of ±0.008 e/Å. Nd atom and its nearby O atoms were AFM coupled, while the Nd atom and its nearby Zn atoms were FM coupled, indicating that the total MM of the doped system had primary contribution from spin-polarized Nd atom(s) and surrounding spin-polarized Zn and O atoms, which has consistency with the findings of atomic and orbital MM analysis.

## 3. Computational Methodology and Models

The energy band structure, electronic states’ density, optical properties, charge density difference (CDD), Mulliken charge population, bond population, and magnetic properties were computed through the CASTEP module [43] of Material Studio 2020. CASTEP is utilizing the first-principles calculation method under the framework of DFT. The norm-conserving pseudopotentials method [44] was utilized for describing the interaction between atomic real and valence electrons, with the generalized gradient approximation Perdew–Burke–Ernzerhof (GGA-PBE) exchange-correlation potential and screened Coulomb potential. The outer valence electrons chosen for the calculation were from configurations of Zn-3d^10^4s^2^, O-2s^2^2p^4^, and Nd-4f^4^5s^2^5p^6^6s^2^, and we treated the rest of the orbital electrons as core electrons. When calculating energy, electrons were treated using spin polarization. The following computational parameters were set for ensuring convergence: the energy tolerance of 5 × 10^−6^ eV/atom and the convergence criteria of 0.01 eV/Å maximum force, 5 × 10^−4^ Å displacement, and 0.02 GPa stress. An energy cut-off of 800 eV and a *k*-point mesh of 4 × 4 × 2 were chosen to balance calculation speed and accuracy. Because the geometrical configuration of Nd plays a substantial role in investigating the magnetic behaviors of Nd-doped ZnO systems, it is important to grasp the realistic geometries of the doped structures. By using the linear response method [45], we calculated the value of U and verified its validity. When U_d-Zn_ = 10.0 eV, U_p-O_ = 7.0 eV, and U_f-Nd_ = 6.0 eV, the computed lattice parameters of a = 3.269 Å, c = 5.272 Å, and bandgap = 3.34 eV were consistent with the experimental results (a = 3.248 Å, c = 5.220 Å [22], and bandgap = 3.3 eV [25]). In previous work, the Nd-doped ZnO with different concentrations (<10%) were fabricated, and modifications were studied. The experiment analysis indicated that the Nd carbonate phase increased with increasing Nd concentration in the ZnO lattice [19,27,28]. In the present work, we estimated the percentage of Nd doping in ZnO, which was referred to in the previous experiment. The super-cellular models used in our calculation are presented in Figure 9, wherein a Zn_16_O_16_ (2 × 2 × 2) supercell comprises 16 O and 16 Zn, a Zn_15_Nd_1_O_16_ (2 × 2 × 2) supercell has 16 O, 15 Zn, and 1 Nd, and a Zn_14_Nd_2_O_16_ (2 × 2 × 2) supercell consists of 16 O, 14 Zn, and 2 Nd. We designed four different configurations for Zn_14_Nd_2_O_16_, and the Nd sites in Zn_14_Nd_2_O_16_ are shown in Figure 9c–f.

## 4. Conclusions

Based on DFT, the electronic, optical, and magnetic properties of Nd-doped ZnO were investigated by the GGA + U method with spin polarization conditions. The findings of the electronic structure revealed that the cell parameters of Nd-doped ZnO systems became larger, the volume became larger, the formation energy was less than zero, the bandgap of Nd-doped ZnO decreased, the impurity energy level introduced high-concentration carriers and degenerates, resulting in the formation of an N-type degenerate semiconductor. Fermi level entered the CB, presenting the features of N-type degenerate semiconductors. The calculation findings of optical properties showed that the impurity energy level and Fermi level generated in the forbidden band of Nd-doped ZnO partially overlapped, and the trap effect was significant, which effectively prevented the recombination of electron–hole pairs. The photogenerated electron transition was completed in a multistage process. In the visible region, the absorption spectrum was obviously redshifted, and its intensity was significantly enhanced, which effectively improved the photocatalytic performance. In the strongest absorption region, with the rise of doping concentration, the absorption spectrum underwent a redshift, expanding the infrared absorption range. The calculation results of magnetic properties indicated that Nd-doped ZnO systems were FM, and the coupling was FM. Magnetism was primarily caused by strong spd-f hybridization due to the magnetic exchange between the Nd-4f state and O-2p, Zn-4s, and Zn-3p states. Therefore, it could be a preferred material for DMS spintronic devices.

## Figures and Tables

**Figure 1 molecules-28-07416-f001:**
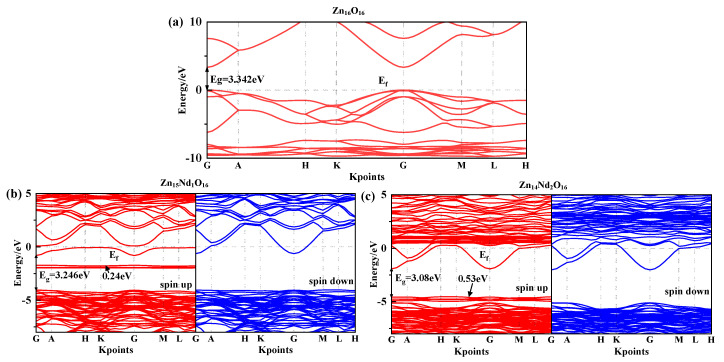
Energy band structure near Fermi surface of (**a**) Zn_16_O_16_, (**b**) Zn_15_Nd_1_O_16_, and (**c**) Zn_14_Nd_2_O_16_.

**Figure 2 molecules-28-07416-f002:**
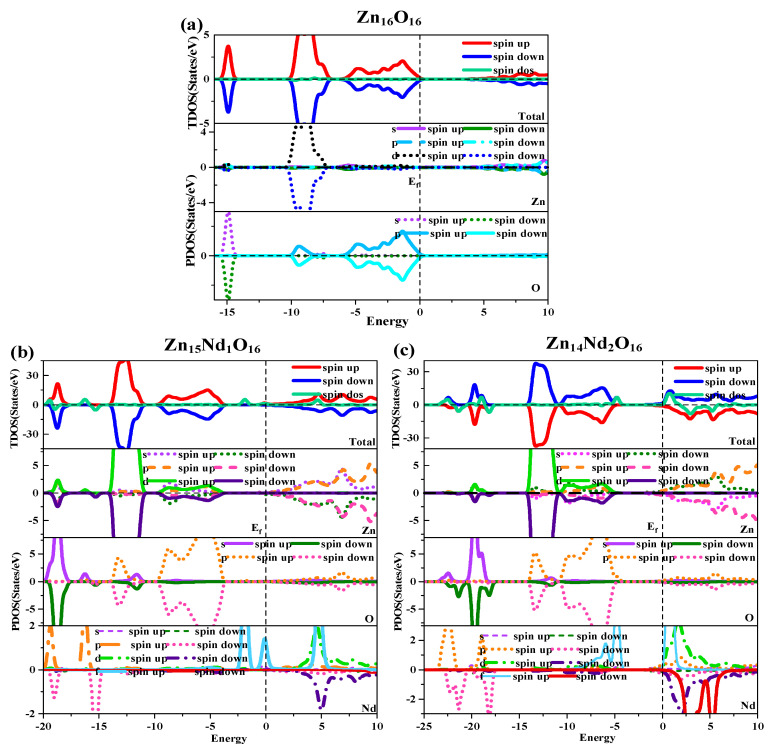
Density of states (DOS) of (**a**) Zn_16_O_16_, (**b**) Zn_15_Nd_1_O_16_, and (**c**) Zn_14_Nd_2_O_16_.

**Figure 3 molecules-28-07416-f003:**
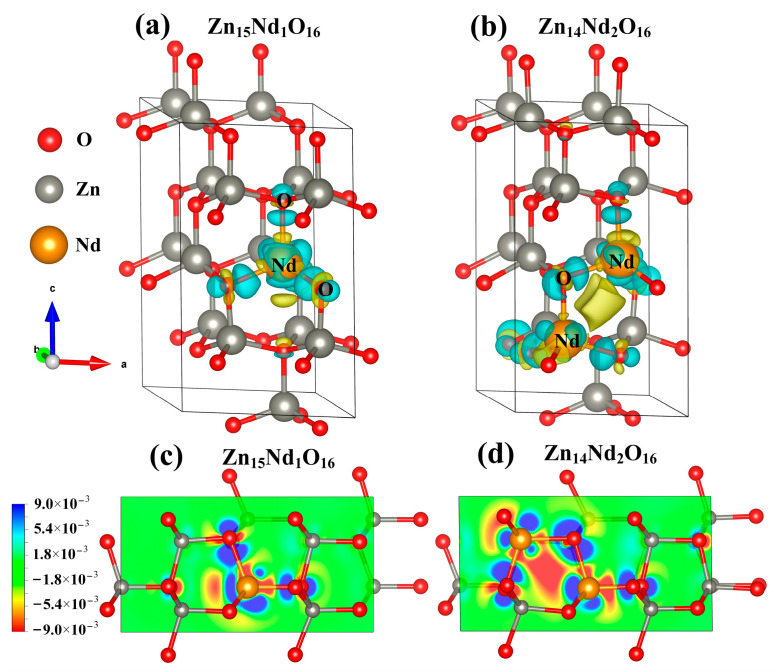
The charge density difference maps of (**a**) Zn_15_Nd_1_O_16_ and (**b**) Zn_14_Nd_2_O_16_. The 2D charge density difference of (**c**) Zn_15_Nd_1_O_16_ and (**d**) Zn_14_Nd_2_O_16_.

**Figure 4 molecules-28-07416-f004:**
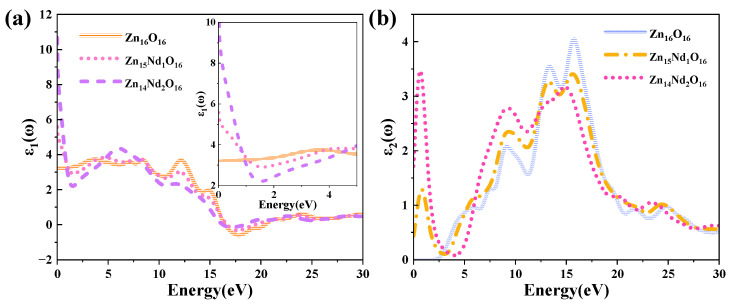
(**a**) Real part and (**b**) imaginary part of the dielectric function.

**Figure 5 molecules-28-07416-f005:**
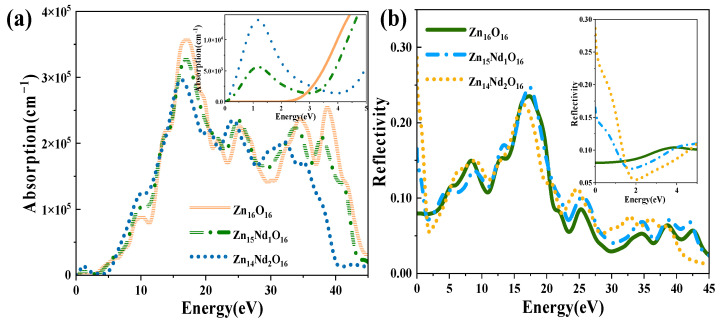
(**a**) Absorption spectrum. (**b**) Reflectivity.

**Figure 6 molecules-28-07416-f006:**
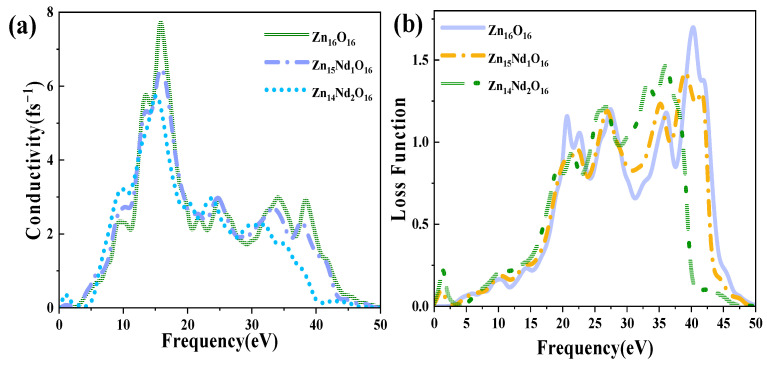
(**a**) Photoconductivity. (**b**) Energy loss function.

**Figure 7 molecules-28-07416-f007:**
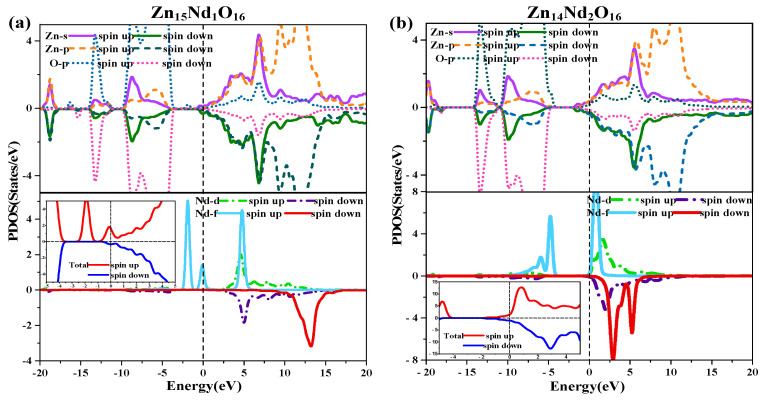
Spin wave state densities of (**a**) Zn_15_Nd_1_O_16_ and (**b**) Zn_14_Nd_2_O_16_.

**Figure 8 molecules-28-07416-f008:**
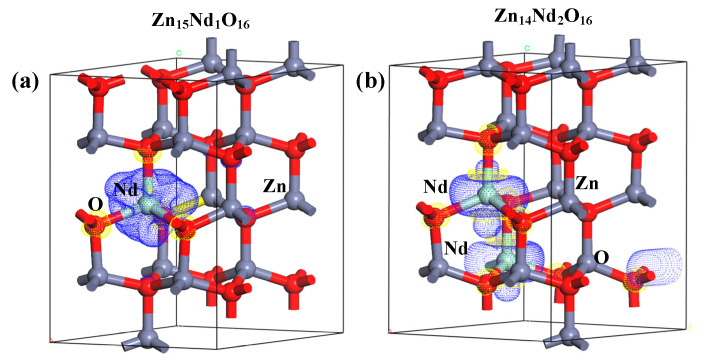
Net spin density distributions of (**a**) Zn_15_Nd_1_O_16_ and (**b**) Zn_14_Nd_2_O_16_.

**Figure 9 molecules-28-07416-f009:**
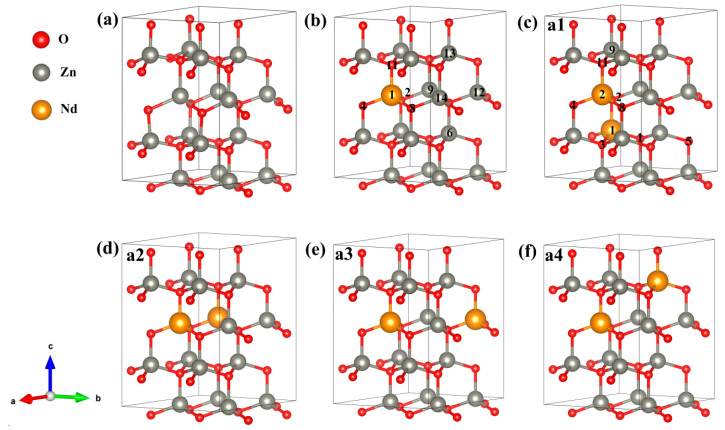
Calculation model of (**a**) Zn_16_O_16_, (**b**) Zn_15_Nd_1_O_16_, (**c**) Zn_14_Nd_2_O_16_
^a1^, (**d**) Zn_14_Nd_2_O_16_
^a2^, (**e**) Zn_14_Nd_2_O_16_
^a3^, and (**f**) Zn_14_Nd_2_O_16_
^a4^. Annotation: a1–a4 represents four different configurations for Zn_14_Nd_2_O_16_; The axis labels a, b, and c for the campass represent coordinates x, y, and z; The numbers in the b,c refers to the atomic positions of Nd and O.

**Table 1 molecules-28-07416-t001:** The lattice parameters of i-ZnO were calculated by this work and compared to the experimental and DFT structure convergence results.

Model	a/Å	c/Å	c/a
This work	3.269	5.272	1.613
Experimental [22]	3.248	5.220	1.607
DFT [8]	3.20	5.15	1.609

**Table 2 molecules-28-07416-t002:** Cell parameters and formation energy of Nd-doped ZnO systems after structural optimization.

Model	a/Å	c/Å	c/a	α	β	γ	V/Å^3^	*E*_f_/eV
Zn_15_Nd_1_O_16_	6.544	10.556	1.613	90.00°	90.00°	120.00°	391.442	−1.83
Zn_14_Nd_2_O_16_ ^a1^	6.625	10.668	1.610	90.00°	90.00°	120.00°	405.471	−3.63
Zn_14_Nd_2_O_16_ ^a2^	6.670	10.643	1.596	89.36°	90.00°	120.38°	403.919	−3.60
Zn_14_Nd_2_O_16_ ^a3^	6.594	10.645	1.614	90.66°	89.34°	119.21°	403.918	−3.60
Zn_14_Nd_2_O_16_ ^a4^	6.612	10.674	1.614	90.00°	89.43°	120.09°	404.854	−3.59

Annotation: ^a1–a4^ represents four different configurations for Zn_14_Nd_2_O_16_.

**Table 3 molecules-28-07416-t003:** Mulliken charge distribution of i-ZnO and Nd-doped ZnO systems.

Model	Atom	S	p	d	f	Total/eV	Net Charge/e
Zn_16_O_16_	Zn (SU)	0.20	0.34	4.99	0.00	5.54	0.93
	Zn (SD)	0.20	0.34	4.99	0.00	5.54	-
	O (SU)	0.92	2.55	0.00	0.00	3.46	−0.93
	O (SD)	0.92	2.55	0.00	0.00	3.46	-
Zn_15_Nd_1_O_16_	Zn (SU)	0.20	0.38	4.99	0.00	5.57	0.88
	Zn (SD)	0.20	0.37	4.99	0.00	5.55	-
	O (SU)	0.90	2.54	0.00	0.00	3.44	−0.91
	O (SD)	0.91	2.56	0.00	0.00	3.46	-
	Nd (SU)	1.21	2.90	0.56	3.70	8.37	1.05
	Nd (SD)	1.14	2.90	0.50	0.03	4.58	-
Zn_14_Nd_2_O_16_	Zn (SU)	0.19	0.35	4.99	0.00	5.53	0.92
	Zn (SD)	0.20	0.35	4.99	0.00	5.54	-
	O (SU)	0.91	2.51	0.00	0.00	3.42	−0.88
	O (SD)	0.91	2.54	0.00	0.00	3.46	-
	Nd_1_ (SU)	1.29	3.02	0.71	3.01	8.03	1.01
	Nd_1_ (SD)	1.26	3.01	0.64	0.05	4.97	-
	Nd_2_ (SU)	1.31	3.05	0.72	3.01	8.10	0.87
	Nd_2_ (SD)	1.29	3.04	0.64	0.05	5.03	-

**Table 4 molecules-28-07416-t004:** Population value and bond length of inherent ZnO and Nd-doped ZnO systems.

Model	Bond	Population Value	Bond Length/Å
Zn_16_O_16_	Zn-O_max_	1.08	1.966
	Zn-O_min_	0.12	1.975
Zn_15_Nd_1_O_16_	Zn-O_max_	0.35	1.985
	Zn-O_min_	0.40	1.967
	Nd-O(//c)	0.22	2.238
	Nd-O(⊥c)	0.34	2.191
Zn_14_Nd_2_O_16_	Zn-O_max_	0.33	1.982
	Zn-O_min_	0.41	1.985
	Nd_1_-O(//c)	0.27	2.260
	Nd_1_-O(⊥c)	0.43	2.195
	Nd_2_-O(//c)	0.35	2.230
	Nd_2_-O(⊥c)	0.46	2.223

**Table 5 molecules-28-07416-t005:** Magnetic coupling modes and total magnetic moment of intrinsic and Nd-doped ZnO systems.

Model	∑M	∑|M|	Coupling Mode	Total Magnetic Moment (µB)
Zn_15_Nd_1_O_16_	3.75	3.92	FM	3.75
Zn_14_Nd_2_O_16_	5.97	6.29	FM	5.97

Annotation: ∑M represents the result of spin-polarized DOS integration. ∑|M| denotes the total absolute value of the spin-polarized DOS integral.

**Table 6 molecules-28-07416-t006:** Atomic magnetic moment and orbital magnetic moment of Nd-doped ZnO systems.

Model	Atoms	Magnetic Moment (µB)	Orbital Magnetic Moment (µB)			
			s	p	d	f
Zn_15_Nd_1_O_16_	O_2_	−0.03	−0.01	−0.02		
	O_4_	−0.03	−0.01	−0.02		
	O_8_	−0.03	−0.01	−0.02		
	O_11_	−0.01	0.00	−0.01		
	Zn_6_	−0.02	–0.02	0.00	0.00	
	Zn_9_	0.01	0.00	0.01	0.00	
	Zn_12_	0.01	0.00	0.01	0.00	
	Zn_13_	−0.01	−0.01	0.00	0.00	
	Zn_14_	0.01	0.00	0.01	0.00	
	Nd	3.79	0.07	0.00	0.06	3.67
Zn_14_Nd_2_O_16_	O_1_	−0.02	0.00	−0.02		
	O_2_	−0.03	0.00	−0.03		
	O_3_	−0.02	−0.01	−0.01		
	O_4_	−0.02	−0.01	−0.01		
	O_5_	−0.02	0.00	−0.02		
	O_8_	−0.02	−0.01	−0.01		
	O_11_	−0.02	0.00	−0.02		
	Zn_9_	−0.01	−0.01	0.00	0.00	
	Nd_1_	3.06	0.03	0.01	0.07	2.96
	Nd_2_	3.08	0.02	0.01	0.08	2.96

## Data Availability

All data are presented in the form of charts in the article.

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
