# Peer review of "Impact of Nd Doping on Electronic, Optical, and Magnetic Properties of ZnO: A GGA + U Study"

_molecules, 2023, doi:10.3390/molecules28217416_

Round 1
Reviewer 1 Report
Comments and Suggestions for Authors
The paper presents numerical investigation of numerous physical properties of zinc oxide doped with neodymium. The analysis focuses on bandgap narrowing, optical properties, and magnetism.
The work employs state-of-the-art computational methods for rare-earth compounds with electron correlations and reports several interesting new results, including the doping-related redshift (and enhanced intensity) in the absorption spectrum and ferromagnetic coupling due to exchange interaction.
Prior to the publication, I suggest the following minor points are considered:
1) Bandgap for an undoped system is compared with the experimental Ref.[18]
Is a similar comparison possible for doped systems?
2) How magnetic properties (of doped systems) can be affected by external pressure?
3) In the case of two Nd atoms per cell: why the two particular sites are chosen? Do different (metastable?) configurations correspond to much higher ground-state energy?
In conclusion, I recommend the paper for publication after optional revisions listed above.
Author Response
Dear Reviewer:
Please check the attachment,thank you!

Reviewer 2 Report
Comments and Suggestions for Authors
The authors calculated the electronic, optical, and magnetic properties of Nd-doped ZnO systems using the DFT/GGA+U method. This work has some merit, however it should be improved.
1/ In line 33 of the introduction, the authors speak about applications of ZnO materials. They should add more references. They can add this recent one (Enhancing the electrical conductivity and the dielectric features of ZnO nanoparticles through Co doping effect for energy storage applications, DOI: 10.1007/s10854-022-09470-5).
Additionally, from line 35 they speak about rare earth elements characteristics. They should add references to reinforce this.
2/ In figures 1b and c, for the structure optimisation, how the authors can estimate the percentage of Nd doping in ZnO, especially to avoid secondary phase (before its solubility limit).
3/ ZnO is kown by its wurtzite structure having two lattice parameters (a and c), but from table 1, there are three parameters for the optimized structure.
T4/ The authors should validate their theoretical results by comparing them with experimental results, especially for optical and magnetic results.
Comments on the Quality of English LanguageMinor editing of English language are required.
Author Response

(The authors gave the same response as above.)

Reviewer 3 Report
Comments and Suggestions for Authors
The fusion of electronic, optical, and magnetic properties is pivotal in shaping the landscape of modern technology. The manuscript employs the DFT/GGA+U method to the Nd-doped ZnO systems.
The study reveals a multifaceted transformation induced by Nd dopants: lattice parameter expansion, negative formation energy, and bandgap narrowing.
A distinctive facet of this study lies in the revelation of the material's magnetic properties. The Nd-doped ZnO system unveils its ferromagnetic nature, attributed to the interactions between Nd-4f state and O-2p, Zn-4s, and Zn-3p states, fostering strong spd-f hybridization. These findings suggest Nd-doped ZnO as a prospective candidate for spintronics applications.
The manuscript is quite close to authors own recent paper:
Wu, Q.; Zhang, B.; Liu, G.; Ning, J.; Shao, T.; Zhang, F.; Xue, S. Study of La Doping on the Electronic Structure and Magneto-Optical Properties of ZnO by GGA+U Method. Crystals 2023, 13, 754. https://doi.org/10.3390/cryst13050754
There are few moments to be addressed before possible publication:
1. Ionic radii provided in the paper lack proper reference.
2. I would recommend a better comparison of calculated and experimental (literature) results. For instance, how close are Nd concentrations in the Zn14NdO16 model to those achieved in experiment?
3. Absence of formation energy for Zn16O16 model doesn't look good. If authors use it as a reference for doped configuration, a zero value could be there.
4. Authors use GGA-U method for band structure calculations. How close the band gap is to the experimental values or those obtained with hybrid potentials?
After correcting these moments the manuscript could be published in the journal.
Author Response

(The authors gave the same response as above.)

Reviewer 4 Report
Comments and Suggestions for Authors
See the attached file.

Sometimes English is too primitive, sometimes incorrect; in any case, the revised text needs a lot of corrections and shoud be checked, preferably, by a native speaker.
Author Response

(The authors gave the same response as above.)

Round 2
Reviewer 2 Report
Comments and Suggestions for Authors
The authors corrected and improved their manuscript as requested. This work can be considered for publication in Molecules.
Reviewer 4 Report
Comments and Suggestions for Authors
Practically all my concerns were answered, so, I can support the acceptance of the revised version.